# Insights into the Rising Threat of Carbapenem-Resistant Enterobacterales and *Pseudomonas aeruginosa* Epidemic Infections in Eastern Europe: A Systematic Literature Review

**DOI:** 10.3390/antibiotics13100978

**Published:** 2024-10-17

**Authors:** Michal Piotrowski, Irina Alekseeva, Urs Arnet, Emre Yücel

**Affiliations:** 1Proper Medical Writing Sp. z o.o., Panieńska 9/12, 03-704 Warsaw, Poland; michal.piotrowski@propermedicalwriting.com; 2Merck Sharp & Dohme, Dubai Healthcare City, Bldg #39, Dubai 2096, United Arab Emirates; irina.alekseeva@merck.com; 3MSD Innovation GmbH, The Circle 66, 8058 Zurich, Switzerland; urs.arnet@msd.com; 4Merck & Co., Inc., Rahway, NJ 07065, USA

**Keywords:** Enterobacterales, *Pseudomonas aeruginosa*, carbapenem resistance, Eastern Europe, systematic literature review, imipenem/relebactam, ceftazidime/avibactam, ceftolozane/tazobactam

## Abstract

Background: Antimicrobial resistance is a major global public health challenge, particularly with the rise of carbapenem-resistant Enterobacterales (CRE) and *Pseudomonas aeruginosa* (CRPA). This study aimed to describe the characteristics of CRE and CRPA infections in Eastern Europe, focusing on Bulgaria, Croatia, Czechia, Greece, Hungary, Poland, Romania, Serbia, Slovakia, and Slovenia. Methods: Following MOOSE and PRISMA guidelines, a systematic literature review of articles published between 1 November 2017 and 1 November 2023 was conducted using the MEDLINE, Embase, Web of Science, CDSR, DARE, and CENTRAL databases. The search strategy used a combination of free text and subject headings to gather pertinent literature regarding the incidence and treatment patterns of CRE and CRPA infections. A total of 104 studies focusing on infections in both children and adults were included in this review. Results: This review revealed a significant prevalence of carbapenem-resistant Gram-negative isolates and underscored the effectiveness of imipenem/relebactam and ceftazidime/avibactam (CAZ/AVI) against Klebsiella pneumoniae carbapenemase-producing Enterobacterales and of ceftolozane/tazobactam, imipenem/relebactam and ceftazidime/avibactam against non-metallo-β-lactamase-producing CRPA strains. Conclusions: This study highlights the urgent need for comprehensive measures to combat the escalating threat of CRE and CRPA infections in Eastern European countries. At the same time, it shows the activity of the standard of care and new antimicrobials against carbapenem-resistant Gram-negative pathogens in Eastern Europe. Clinical real-world data on the treatment of carbapenem-resistant infections in Eastern Europe are needed.

## 1. Introduction

Antimicrobial resistance has led to a global public health crisis by rendering antibiotics ineffective against bacterial infections [1]. If left unchecked, it could contribute to 10 million annual deaths worldwide by 2050, including 390,000 deaths in Europe [2]. In 2019, 133,000 deaths were attributed to bacterial antimicrobial resistance in the World Health Organization (WHO) European region, with Eastern Europe showing the highest mortality rate of 19.9 per 100,000 population [3]. Antimicrobial resistance also imposes a financial burden, estimated to be at least EUR 1.5 billion annually in Europe [4]. Treatment costs for resistant Gram-negative bacterial infections are approximately 29% higher than those for infections caused by susceptible strains, and they are associated with longer hospital stays and higher mortality rates [5,6].

In response to increasing antimicrobial resistance, the WHO, Infectious Diseases Society of America, and Centers for Disease Control and Prevention have identified critical pathogens, including carbapenem-resistant isolates. The most relevant Gram-negative bacteria discussed include carbapenem-resistant Enterobacterales (CRE) (*Klebsiella pneumoniae* and *Escherichia coli*), *Pseudomonas aeruginosa*, and *Acinetobacter baumannii* [7,8,9].

Gram-negative bacterial infections are commonly treated with β-lactam antibiotics, but resistance mechanisms such as β-lactamase production have evolved to encompass broader-spectrum antibiotics, including carbapenems [10,11]. *P. aeruginosa* develops resistance to antibiotics not only through the production of carbapenemases but also through other mechanisms such as the increased expression of efflux pumps, the reduced permeability of the outer membrane, and the overproduction of AmpC β-lactamase. Nonenzymatic mechanisms, specifically efflux pumps and porin deficiency, along with the overproduction of AmpC, are responsible for approximately 85% to 95% of resistance in *Pseudomonas*. In Eastern Europe, the prevalence of carbapenemase-producing *P. aeruginosa* strains is estimated at 5% to 15%. As carbapenems are last-line broad-spectrum antibiotics, different antimicrobial treatments are needed to control Gram-negative infections, addressing both carbapenemase-related and nonenzymatic mechanisms of bacterial resistance [10]. In European countries, 1.3 patients per 10,000 hospital admissions were infected with CRE in the years 2013 and 2014, with the highest infection rates in Mediterranean and Balkan countries [12]. Carbapenem-resistant *K. pneumoniae* has been reported increasingly in the Balkan region, with the highest number of cases observed in Greece and Romania [13].

In 2021, significant differences in the prevalence of carbapenem-resistant *P. aeruginosa* (CRPA) were reported, with the lowest percentage of invasive CRPA isolates in Denmark (6.8%), Finland (4.2%), the Netherlands (5.2%), and Norway (6.8%) and the highest in Belarus (60.2%), Romania (45.9%), Serbia (62.6%), and Ukraine (78.0%) [14].

In Eastern Europe, treatment options for carbapenem-resistant Gram-negative infections include old antibiotics (e.g., fosfomycin, colistin, tigecycline, and aminoglycosides), new antibiotics (e.g., plazomicin, eravacycline, and cefiderocol), new β-lactam/β-lactamase inhibitor combinations (e.g., ceftazidime/avibactam (CAZ/AVI), meropenem/vaborbactam [MERO/VAB], and imipenem/relebactam [IMI/REL] for CRE; ceftolozane/tazobactam [C/T] and ceftazidime/avibactam [CAZ/AVI] for CRPA), and combination therapies (e.g., double-carbapenem therapy, new β-lactam/β-lactamase inhibitor combinations with colistin, fosfomycin, aminoglycosides, and fluoroquinolones) [15].

The treatment of infections caused by carbapenem-resistant bacteria depends on the type of enzyme produced: for Enterobacterales producing OXA-48-like enzymes, CAZ/AVI is used; for *K. pneumoniae* carbapenemase (KPC), treatments include MERO/VAB, IMI/REL, and CAZ/AVI; and for metallo-β-lactamases (MBLs), a combination of CAZ/AVI with aztreonam or cefiderocol is recommended [11,16]. For patients with severe infections caused by non-MBL CRPA, C/T is suggested as a preferred treatment. There is limited evidence on the real-world effectiveness of IMI/REL, cefiderocol, and CAZ/AVI in the treatment of CRE/KPC-associated infections (16). Conventional and less novel antibiotics (e.g., colistin and fosfomycin) may still be used in patients with these infections (16).

In 2020, the European Medicines Agency (EMA) approved IMI/REL for the treatment of adults with hospital-acquired pneumonia, including ventilator-associated pneumonia, bacteremia associated with or suspected to be associated with hospital-acquired or ventilator-associated pneumonia, and infections due to aerobic Gram-negative organisms with limited treatment options [17]. Moreover, the EMA approved C/T for complicated intra-abdominal and urinary tract infections in 2014, for hospital-acquired or ventilator-associated pneumonia in 2019, and for pediatric use in complicated intra-abdominal and urinary tract infections in 2022. Recent antimicrobial susceptibility data showed that C/T and IMI/REL are important treatment options for patients in Czechia, Hungary, Poland, and Greece with infections caused by resistant Gram-negative pathogens, including Enterobacterales and *P. aeruginosa* [18,19].

This study aimed to summarize the available evidence on the clinical and economic disease burden and treatment patterns of carbapenem-resistant Gram-negative infections among adults and children, with a focus on CRE and CRPA in Eastern European countries (i.e., Bulgaria, Croatia, Czechia, Greece, Hungary, Poland, Romania, Serbia, Slovenia, and Slovakia). Here, we report findings on clinical evidence.

## 2. Results

Of the 758 citations initially identified, 147 full-text articles were retrieved as potentially eligible for inclusion. After a thorough review, a total of 104 studies were included in the systematic literature review (Figure 1). Detailed information on the publications is provided in the Appendix A. Among the included articles, 27 were from Poland [18,20,21,22,23,24,25,26,27,28,29,30,31,32,33,34,35,36,37,38,39,40,41,42,43,44,45], 24 from Greece [46,47,48,49,50,51,52,53,54,55,56,57,58,59,60,61,62,63,64,65,66,67,68,69], 16 from Romania [70,71,72,73,74,75,76,77,78,79,80,81,82,83,84,85], 11 from Croatia [86,87,88,89,90,91,92,93,94,95,96], 9 from Serbia [97,98,99,100,101,102,103,104,105], 7 from Bulgaria [106,107,108,109,110,111,112], 5 from Czechia [18,113,114,115,116], 4 from Hungary [18,117,118,119], and 1 from Slovakia [120]. There were no publications from Slovenia that met the inclusion criteria.

### 2.1. Diagnostic Methods

The most common bacterial identification methods used in the included studies were the automated VITEK system and MALDI coupled to time-of-flight mass spectrometry (MALDI-TOF MS). Antimicrobial susceptibility testing methods included the disk diffusion method, broth microdilution, the VITEK or Phoenix system, and E-tests, with a similar distribution across studies. For carbapenemase detection, the modified Hodge test, disk synergy test, Carba NP test, and carbapenem inactivation method (CIM) were used at similar rates across studies.

### 2.2. Epidemiology and Antibiotic Susceptibility of CRE

Among the analyzed studies, the highest number of CRE cases was reported for Greece, Poland, Romania, and Serbia. KPC carbapenemases were most prevalent in Greece and Bulgaria; New Delhi MBLs (NDM) in Czechia and Poland; Verona integron-encoded MBLs (VIM) in Hungary and Croatia; and OXA-48 in Serbia, Poland, and Romania (Table 1).

After excluding studies that analyzed 100% strains resistant to carbapenems, we calculated the rate of CRE as follows: Bulgaria 83.7%, Croatia 47.9%, Greece 49.4%, Hungary 1.8%, Poland 9.4%, Romania 18.0%, and Serbia 62.3%. For Czechia and Slovakia, a recalculation was not possible, as all studies from these countries included only strains resistant to carbapenems (Figure 2).

We identified eight studies [22,47,52,58,59,75,88,105] that evaluated the activity of both IMI/REL and CAZ/AVI against CRE. The resistance rate for IMI/REL ranged from 7.3% to 100%, while that for CAZ/AVI ranged from 0.3% to 100%. The lowest resistance rates to both antibiotics were observed for *K. pneumoniae* in a study by Galani et al. [52], although it should be noted that 94% of the strains tested were KPC producers. In a study by Bedenić et al. [88], all 10 strains tested, including 8 *K. pneumoniae* strains, were resistant to IMI/REL, and 50% were resistant to CAZ/AVI. All these strains were OXA-48 producers, and 50% also produced NDM. In a study by Biedrzyca et al. [22], *K. pneumoniae* strains also showed high resistance rates to both antibiotics, with all tested strains being VIM producers. Detailed data are presented in Table 2.

An additional 14 studies assessed only the activity of CAZ/AVI [22,34,35,44,45,48,49,54,61,69,79,106,107,112]. The resistance rate for CAZ/AVI in these studies ranged from 0.7% to 100%. In the three studies reporting a resistance rate of 100%, all strains produced MBLs [54,106,107].

### 2.3. Carbapenem-Resistant P. aeruginosa Susceptibility to Antibiotics and Resistance Mechanisms

Among the analyzed studies, the highest numbers of CRPA cases were reported in Poland, Hungary, and Greece. High rates of MBL producers were found in Bulgaria, Croatia, Czechia, Hungary, and Poland (Table 3).

After excluding studies that analyzed 100% strains resistant to carbapenems, we calculated the rate of CRPA as follows: Croatia 70.4%, Czechia 47.7%, Greece 31.2%, Hungary 19.1%, Poland 17.6%, Romania 46.4%, and Serbia 53.4%. For Bulgaria and Slovakia, a recalculation was not possible, as all studies from these countries included only strains resistant to carbapenems (Figure 3).

We identified three studies [45,107,119] that assessed susceptibility to CAZ/AVI and C/T. In a study by Kostyanev et al. [107], all five CRPA strains were resistant to CAZ/AVI and C/T. The included strains were also resistant to other antipseudomonal agents, such as piperacillin/tazobactam, ceftazidime, cefepime, and fluoroquinolones, but susceptible to colistin, and all CRPA produced NDM carbapenemases [107]. O’Neall et al. [119] tested the susceptibility of 65 CRPA strains to CAZ/AVI and C/T. The resistance rate was 98.4% for both antibiotics, with most strains (90.8%) being MBL producers. In a study by Zalas-Wiecek et al. [45], the resistance rates for CAZ/AVI and C/T were 50.5% and 42.8%, respectively, with all tested strains being susceptible to colistin.

## 3. Discussion

Antimicrobial resistance is one of the top 10 global public health challenges. In 2022, the European Commission, together with the European Union Member States, identified it as one of the three highest-priority health threats [122,123]. Numerous studies have linked the use of carbapenems to the development of carbapenem resistance [124,125]. Carbapenems are the third most commonly used class of antibiotics worldwide for the treatment of community-acquired infections in the intensive care unit (10.7%) and the foremost class for managing healthcare-associated infections (21.5%) [126]. In 2022, the most significant increase in antimicrobial resistance within the European Union/European Economic Area was observed for carbapenem-resistant *K. pneumoniae*, as measured by the population-weighted mean percentage increase from 2018, across all bacterial species and antimicrobial group combinations. This escalating trend in resistance, coupled with a marked increase in the estimated incidence of bloodstream infections across the European Union over the same period, raises significant concerns [127].

The objective of our study was to conduct a systematic literature review to outline the characteristics of CRE and carbapenem-resistant *P. aeruginosa* in Eastern European countries, specifically Bulgaria, Croatia, Czechia, Greece, Hungary, Poland, Romania, Serbia, Slovakia, and Slovenia. The economic burden study could not be completed due to an insufficient number of publications for the region. We also aimed to assess treatment patterns, but they were not covered by available data. Instead, we included data on antimicrobial susceptibility to antibiotics.

The European Centre for Disease Prevention and Control and WHO 2023 report highlighted a significantly higher incidence of infections caused by carbapenem-resistant pathogens in Eastern European countries compared with Western Europe in 2021. For example, in Greece, the prevalence of carbapenem-resistant *K. pneumoniae* infections was 73.7% compared with 54.5%, 47.9%, and 46.3% in Romania, Serbia, and Bulgaria, respectively. In contrast, Czechia reported a prevalence of only 1%. Similar trends were observed for CRPA, with higher resistance prevalence in Eastern Europe compared with Western Europe. This alarming increase in carbapenem-resistant Gram-negative strains across most Eastern European countries in 2017–2021 indicates the urgent need for comprehensive measures, including surveillance, antimicrobial stewardship, infection prevention and control, and public health initiatives, to effectively address this growing threat and protect public health.

The methodology of the included studies indicates that the countries analyzed used advanced automated techniques to identify bacterial species. Carbapenemases were detected by different methods such as the Carba NP, CIM, or modified Hodge test. The early detection of carbapenemase producers allows for the timely initiation of appropriate antibiotic therapy, which is particularly important in severe infections. Additionally, it can help hospitals implement necessary infection control measures to prevent the spread of resistant bacteria. O’Neal et al. [119] suggested that the CIM could be useful for the routine testing of carbapenemase activity in Gram-negative organisms. According to the authors, a negative CIM test result for *P. aeruginosa* suggests that C/T could be an effective treatment, even in cases with carbapenem resistance.

Our study showed significant variations in the rates of antimicrobial resistance and the types of resistance-causing enzymes among different countries and bacterial strains. These variations could be caused by various factors and underlying mechanisms, including antimicrobial use, infection control measures, healthcare settings, and bacterial genetic characteristics. Further studies are needed to gain a deeper understanding of these variations and to guide the development of effective strategies to combat antibiotic resistance.

New treatment options have emerged for infections caused by carbapenem-resistant bacteria, with varying effectiveness depending on the organism and specific class of carbapenemases produced. These options include C/T, CAZ/AVI, IMI/REL, and cefiderocol for CRPA, while for CRE, MERO/VAB, CAZ/AVI, IMI/REL, cefiderocol, and other antibiotics are often used in combination therapies (plazomicin, eravacycline, and fosfomycin) [128,129,130,131]. There are few treatment options for infections caused by class B carbapenemase-producing bacteria (new β-lactam/β-lactamase inhibitors in combination with aztreonam, colistin, and cefiderocol). This is a major concern in countries such as Bulgaria, Croatia, Greece, Hungary, Romania, and Serbia, where KPC producers are highly prevalent. Interestingly, a shift from the predominance of KPC-producing bacteria to MBL-producing bacteria was observed in Greece, possibly related to the excessive empirical use of newer antimicrobial agents such as CAZ/AVI [62,69]. In our study, we focused on new antimicrobial agents used for the treatment of infections caused by CRE and CRPA. IMI/REL and CAZ/AVI showed good activities in studies with KPC-producing Enterobacterales. In addition, CAZ/AVI exhibited good activity against OXA-48 producers. C/T and CAZ/AVI were highly effective against non-carbapenemase-producing CRPA strains. In our systematic literature review, two of three studies reported that all strains were resistant to both agents due to MBL production. In the third study, the resistance rates for CAZ/AVI and C/T were 50.5% and 42.8%, respectively [45,107,119].

Containing the spread of carbapenem resistance, particularly in healthcare settings, requires a multifaceted approach. This should be focused on antimicrobial stewardship programs, infection prevention and control, rapid diagnostics, the education and training of healthcare workers, and research into and the development of new antibiotics [132,133,134]. Future clinical studies should focus on the development and evaluation of novel antibiotics specifically targeting CRE and CRPA. Investigating combination therapies that may enhance the efficacy of existing drugs or prevent resistance development is another area of interest. Further studies on resistance mechanisms, including the role of plasmid-mediated genes, could help in identifying new targets for therapeutic intervention. Clinical trials evaluating the effectiveness and safety of emerging treatments, as well as strategies for optimizing antibiotic stewardship, will be essential in combating CRE and CRPA infections.

The limitations of our systematic literature review include differences in methodology between the included studies, the lack of a central laboratory, the lack of published data from clinical studies, and the retrospective design of most studies. Therefore, our results should be interpreted with caution, especially since there was a small number of isolates tested against modern antimicrobials.

## 4. Materials and Methods

### 4.1. Literature Search

This systematic literature review was conducted following the MOOSE and PRISMA guidelines using databases such as MEDLINE, Embase, Web of Science, CDSR, DARE, and CENTRAL. Additional studies were identified by searching reference lists and national journals. The search strategy covered articles published from 1 November 2017 to 1 November 2023 and incorporated a combination of free text searching and subject headings, eliminating the need for multiple synonyms for each term. This approach aimed to identify the most relevant literature on Enterobacterales, *P. aeruginosa*, carbapenem resistance, and the countries within the scope of the systematic literature review. The search strategy is described in detail in the Appendix A. This study was not pre-registered. 

### 4.2. Inclusion Criteria

The primary inclusion criteria comprised publications on bacterial infections or colonization caused by CRE and CRPA in both children and adults from the following countries: Bulgaria, Croatia, Czechia, Greece, Hungary, Poland, Romania, Serbia, Slovakia, and Slovenia. We searched for publications in either English or the local language that reported outcomes related to the incidence or prevalence of CRE and/or CRPA infections and antimicrobial treatment patterns.

Publications describing infections susceptible to carbapenems and those reporting on studies conducted during a hospital infection outbreak were excluded.

### 4.3. Study Selection and Data Extraction

Two investigators (M.P.) searched the databases and exported the results to EndNote. Two independent investigators (K.B-M) screened the data. Conflicting outcomes were resolved by a third (senior) investigator. The identified studies were deduplicated in EndNote before being transferred to Excel and assessed against the inclusion criteria. Study selection was performed in two steps: title and abstract screening followed by full-text screening.

For title and abstract screening, the titles and abstracts of publications identified from the database searches were screened by two investigators to determine their eligibility according to the inclusion criteria. If the relevance of a study was unclear, a senior investigator was consulted. Articles that did not meet the inclusion criteria were excluded from further review, and the reasons for exclusion were recorded.

Full-text screening was performed by two investigators to determine the eligibility of publications using the same inclusion and exclusion criteria as for title and abstract screening. If the relevance of a study was unclear, a senior investigator was consulted.

Relevant publication data were extracted using a predeveloped data extraction form. Data extraction was performed by one investigator, while a senior investigator carefully reviewed the extracted data for accuracy. The extracted data included the title and year of publication, authors, study type, country, site of infection, type of hospital ward, sample size, bacteria type, resistance mechanisms, resistance genes, antibiotics tested, minimum inhibitory concentrations, treatment outcomes, diagnostic methods, infection risk factors, and economic burden.

## 5. Conclusions

In conclusion, IMI/REL and CAZ/AVI demonstrated good comparable activity against KPC-producing Enterobacterales. C/T and CAZ/AVI showed high activity against CR *P. aeruginosa* strains that do not produce MBLs. Our study also showed a significant prevalence of carbapenem-resistant Gram-negative isolates in Eastern European countries. This highlights the urgent need for a multifaceted approach to address this public health challenge. Further research into the mechanisms underlying carbapenem resistance and real-world clinical studies in patients with carbapenem-resistant infections are needed to develop new antimicrobial agents that would be effective against these infections. Moreover, it is necessary to implement and strengthen effective public health strategies, both regionally in Eastern Europe and globally, to reduce the spread of carbapenem-resistant bacteria. Such efforts should be aimed at optimizing antibiotic use, improving infection control measures, and promoting international collaboration in research and public health initiatives.

## Figures and Tables

**Figure 1 antibiotics-13-00978-f001:**
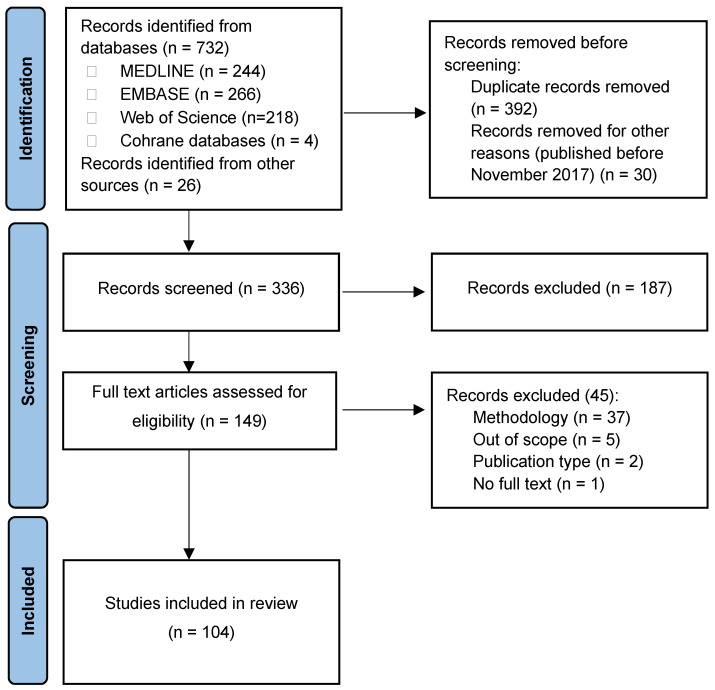
Preferred Reporting Items for Systematic Reviews and Meta-Analyses (PRISMA) flow chart of the study selection process.

**Figure 2 antibiotics-13-00978-f002:**
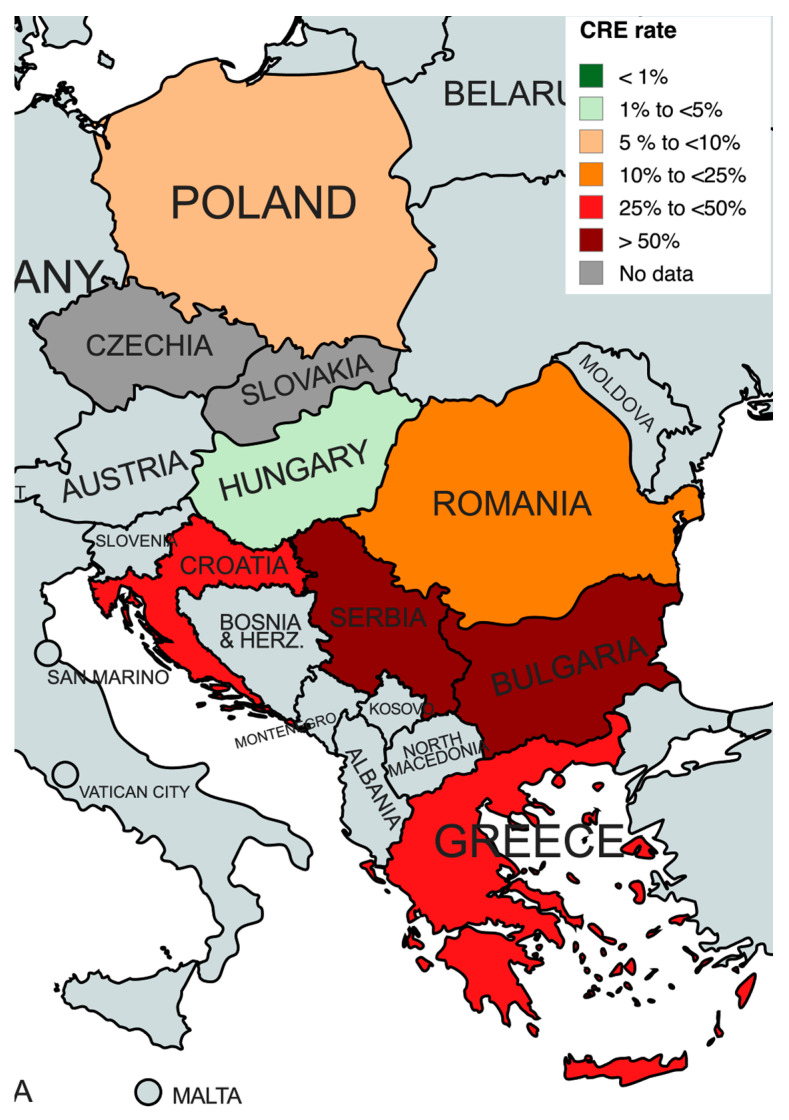
Carbapenem-resistant Enterobacterales rate in analyzed countries. Created with mapchart.net.

**Figure 3 antibiotics-13-00978-f003:**
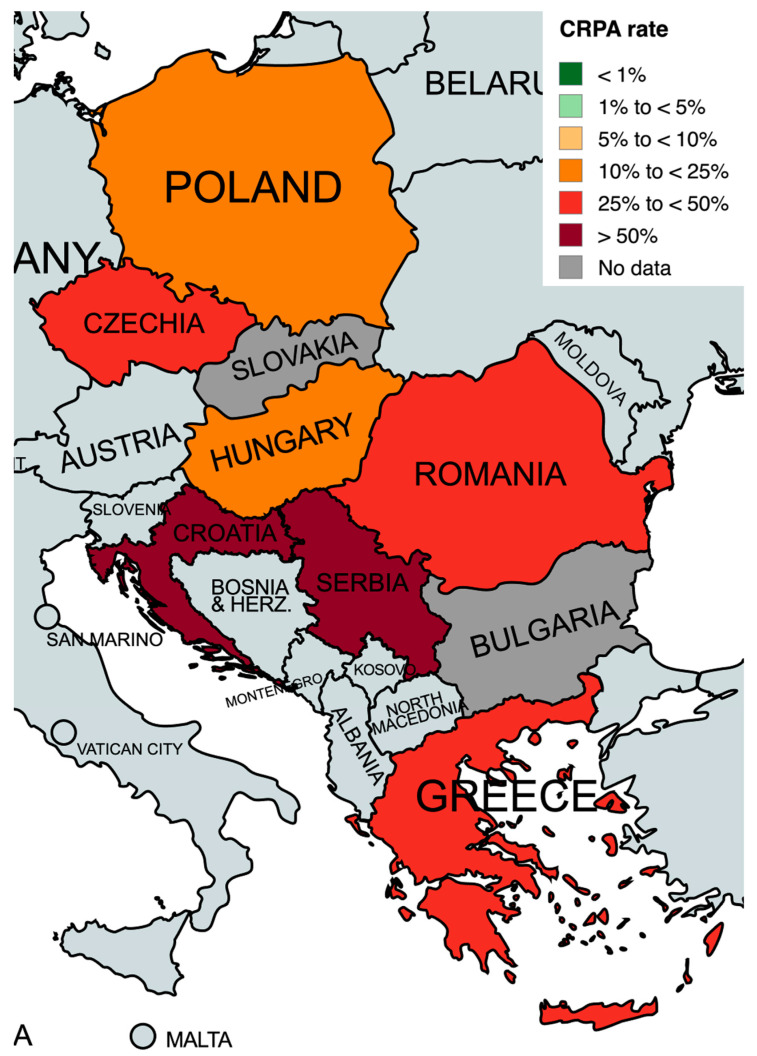
Carbapenem-resistant *P. aeruginosa* rate in analyzed countries. Created with mapchart.net.

**Table 1 antibiotics-13-00978-t001:** Carbapenem-resistant Enterobacterales and resistance mechanisms reported in the included studies (n, %).

Country	No. of Isolates	CR Rate n (%)	Prevalence of KPC Mechanism ^a^ (n/N, %)	Prevalence of MBL Mechanism ^a^ (n/N, %)	Prevalence of OXA-48 Mechanism ^a^ (n/N, %)	References
Bulgaria	188	188 (100)	85/162 (52.5)	NDM: 87/188 (46.3) VIM: 16/150 (10.7)	OXA-48: 3/150 (2)	[106,111]
Croatia	588	583 (99.2)	57/474 (12.0)	VIM: 149/504 (26.6) NDM: 26/428 (6.1)	OXA-48: 83/474 (17.5)	[86,87,88,90,91,92,93,94,96]
Czechia	33	33 (100)	10/33 (30.3)	NDM: 23/33 (69.7)	OXA-48: 4/11 (36.4)	[113,114,116]
Greece	7257	4819 (66.4)	2328/3872 (60.1)	NDM: 417/3469 (12) VIM: 317/3440 (9.2)	OXA-48: 154/3872 (4.0)	[46,47,49,50,51,52,53,54,55,56,57,58,59,60,61,62,65,66,67,68,69]
Hungary	1474	27 (1.8)	-	VIM: 21/27 (77.8) NDM: 2/27 (7.4)	OXA-48: 6/27 (22.2)	[18,118]
Poland	6968	3324 (47.7)	202/743 (27.2)	NDM: 2263/2986 (75.8) VIM: 137/850 (16.1)	OXA-48: 359/743 (48.3)	[20,21,22,23,24,25,27,29,30,32,33,34,35,37,38,39,40,41,43,44,45]
Romania	3396	790 (23.2)	24/220 (10.9)	NDM: 103/268 (38.4) VIM: 7/112 (6.3)	OXA-48: 96/220 (43.6)	[70,71,72,73,74,75,76,77,78,79,80,81,85]
Serbia	2724	703 (25.8)	24/188 (12.8)	NDM: 14/188 (7.4)	OXA-48: 123/188 (65.4)	[97,98,100,105]
Slovakia	14	100 (100)	ND	ND	ND	[120]

a—not every strain tested; CR—carbapenem-resistant; KPC—*K. pneumoniae* carbapenemase; MBL—metallo-β-lactamase; ND—no data; NDM—New Delhi metallo-β-lactamases; VIM—Verona integron-encoded metallo-β-lactamases.

**Table 2 antibiotics-13-00978-t002:** Activity of imipenem/relebactam and ceftazidime/avibactam against Enterobacterales and resistance mechanisms in the included studies.

Country	Pathogen	No. of CR Strains	IMI/REL Resistance (%)	CAZ/AVI Resistance (%)	Resistance Mechanisms (%) ^a^	Reference
Croatia	Enterobacterales	10	100	50	OXA-48: 100 NDM: 50 KPC: 20 VIM: 10	[88]
Greece	*K. pneumoniae*	314	7.3	0.3	KPC: 94 OXA-48: 6	[52]
Greece	*K. pneumoniae*	266	24	20.3	KPC: 75.6 NDM: 11.7 VIM: 5.6 OXA-48: 4.1	[58]
Greece	*K. pneumoniae*	110	34.5	33.7	KPC: 64.6 NDM: 20.9 VIM: 8.2 OXA-48: 2.7	[59]
Greece	Enterobacterales	422	39.8	40.2	KPC: 57.3 MBL: 44 OXA-48: 8.8	[47]
Poland	*Klebsiella* spp.	106	82.6	82.6	VIM: 100	[22]
Romania	*K. pneumoniae*	10	50	20	OXA-48: 40 KPC: 40 NDM: 20	[75]
Serbia	*K. pneumoniae*	143	51.8	25.9	OXA-48: 60 NDM: 8.2 KPC: 16.5	[105]

a—a single strain may have more than one resistance mechanism; CAZ/AVI—ceftazidime/avibactam; CR—carbapenem-resistant; IMI/REL—imipenem/relebactam.

**Table 3 antibiotics-13-00978-t003:** Carbapenem-resistant *P. aeruginosa* resistance mechanisms in the included studies.

Country	No. of Isolates	CR Rate, n (%)	MBL Mechanism Prevalence ^a^, n/N (%)	Other Mechanisms ^a^, n/N (%)	Reference
Bulgaria	80	74 (92.5)	NDM: 5/5 (100)	OXA-50: 3/32 (9.3)	[107,109,110]
Croatia	70	54 (77.1)	VIM: 8/8 (100)	GES: 5/5 (100)	[86,87,91]
Czechia	289	167 (57.9)	VIM: 15/138 (10.9) IMP: 119/138 (86.2)	GES: 4/138 (2.9)	[115]
Greece	1179	376 (31.9)	VIM: 13/39 (33.3) NDM: 9/39 (23.1)	ND	[57,65,67,121]
Hungary	2284	436 (19.1)	VIM: 64/77 (83.1) NDM: 7/65 (10.8)	ND	[18,117,119]
Poland	1555	805 (51.8)	MBL: 486/489 (99.4)	ND	[23,24,26,40,42,43,44,45]
Romania	386	179 (46.4)	ND	ND	[71,77]
Serbia	1111	593 (53.4)	NDM: 31/138 (22.5)	ND	[100,101,102]
Slovakia	16	16 (100)	ND	ND	[120]

a—not every strain tested; CR—carbapenem-resistant; MBL—metallo-β-lactamase; ND—no data.

## Data Availability

The data that support the findings of this study are available on request from the corresponding author.

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
