# Peer review of "Insights into the Rising Threat of Carbapenem-Resistant Enterobacterales and Pseudomonas aeruginosa Epidemic Infections in Eastern Europe: A Systematic Literature Review"

_antibiotics, 2024, doi:10.3390/antibiotics13100978_

Round 1
Reviewer 1 Report
Comments and Suggestions for Authors
This manuscript provides a comprehensive analysis of the prevalence, epidemiology, and antimicrobial resistance of Enterobacterales and Pseudomonas aeruginosa, specifically focusing on Eastern European countries. The authors summarize the current state of carbapenem resistance in these organisms and evaluate the effectiveness of various treatment options, particularly the β-lactam/β-lactamase inhibitor combinations. Overall, this paper provides valuable insights into the epidemic of carbapenem resistance field. have several concerns needed to be addressed as follows:
1. The Percent match of the manuscript is 25% which is pretty high, and needed to be reduced to below 15 percent at least.
2. The authors should consider updating the data to include the most recent antimicrobial susceptibility reports, as resistance patterns can evolve rapidly.
3. Suggest potential areas for future research, such as the evaluation of novel antibiotics and combination therapies for treating carbapenem-resistant infections.
4. the authors should add some figures to illustrate their findings instead of only tables.
5. the authors should consider to expand the discussion to include potential strategies for containing the spread of carbapenem resistance.
Comments on the Quality of English LanguageModerate editing of English language required.
Author Response
Authors: Thank you very much for your time and this revision. Please find below our point-by-point answers. We hope we have improved the work enough to make it suitable for publication
- The Percent match of the manuscript is 25% which is pretty high, and needed to be reduced to below 15 percent at least.
Authors: We are not entirely sure what 'match of the manuscript' means. As for plagiarism, we checked the work using the software Plagiat.pl, and the results showed a WP1 coefficient of 5% and a WP2 coefficient of 0%. The interpretation is as follows: Results are considered to require detailed analysis if WP1 exceeds 50% and WP2 exceeds 5%.
- The authors should consider updating the data to include the most recent antimicrobial susceptibility reports, as resistance patterns can evolve rapidly.
Authors: We agree with this opinion and thank you for the suggestion. However, the data analysis and synthesis took us a long time, and these are the original dates that we included in the protocol of our review. We have planned to update the review every two years.
- Suggest potential areas for future research, such as the evaluation of novel antibiotics and combination therapies for treating carbapenem-resistant infections.
Authors: Thank you for this suggestion. We have added such a paragraph at the end of the discussion.
- the authors should add some figures to illustrate their findings instead of only tables.
Authors: Thank you for this suggestion. We have added maps illustrating CR rates calculated based on the included studies.
5. the authors should consider to expand the discussion to include potential strategies for containing the spread of carbapenem resistance.
Authors: Thank you, we have expanded discussion as suggested.
Reviewer 2 Report
Comments and Suggestions for Authors
I have read with interest the manuscript submitted by Piotrowski et al, since AMR represents a global concern.
Overall, the manuscript is well-written and organized. I have a few comments to be addressed in order to improve the quality of the manuscript.
- Latin bacterial names should be italicized in the abstract as well;
- not all articles containing data about carbapenemases in Eastern Europe were included; for example DOI: 10.3390/antibiotics11050548
(Out of 354 isolates, 25 (7%) were CRE, with 19 isolates being carbapenemase producers, as determined by the NG-Test Carba 5 multiplex immunoassay. Of these, 17 isolates were K. pneumoniae, one was E. coli (NDM), and one was Enterobacter cloacae (OXA-48). The most commonly identified carbapenemase was OXA-48 (8 isolates), followed by NDM (7 isolates), KPC (3 isolates), and VIM (1 isolate).)
- it would be interesting to highlight CRE trends during the analyzed period, and, if possible, include the percentages of carbapenemases to show if there are any shifts.
- for increased addressability, it would be great to include a figure comparing the results from each country analyzed in the result section (chart of even better - map).
- the reference list is appropriate and properly edited.
Best regards,
Comments on the Quality of English Languageminor
Author Response
Authors: Thank you very much for this revision and for your kind words about our work. Please find below our point-by-point answers. We hope we have improved the work enough to make it suitable for publication
- Latin bacterial names should be italicized in the abstract as well;
Authors: Thank you, we have corrected it.
- not all articles containing data about carbapenemases in Eastern Europe were included; for example DOI: 10.3390/antibiotics11050548
(Out of 354 isolates, 25 (7%) were CRE, with 19 isolates being carbapenemase producers, as determined by the NG-Test Carba 5 multiplex immunoassay. Of these, 17 isolates were K. pneumoniae, one was E. coli (NDM), and one was Enterobacter cloacae (OXA-48). The most commonly identified carbapenemase was OXA-48 (8 isolates), followed by NDM (7 isolates), KPC (3 isolates), and VIM (1 isolate).
Authors: Thank you. We're not sure how we missed this study, but we have now added it to our analysis.
- it would be interesting to highlight CRE trends during the analyzed period, and, if possible, include the percentages of carbapenemases to show if there are any shifts.
Authors: Thank you for this suggestion. Anyway, we are unable to demonstrate a clear trend based on the selected published data, as the participation of different sites varies over time. The inconsistency in data collection from various locations makes it difficult to draw definitive conclusions about trends.
- for increased addressability, it would be great to include a figure comparing the results from each country analyzed in the result section (chart of even better - map).
Authors: Thank you for this suggestion. We have added maps illustrating CR rates calculated based on the included studies.
- the reference list is appropriate and properly edited.
Authors: Thank you.